# Sociodemographic, labour market marginalisation and medical characteristics as risk factors for reinfarction and mortality within 1 year after a first acute myocardial infarction: a register-based cohort study of a working age population in Sweden

Mo Wang [1], Marjan Vaez,[1] Thomas Ernst Dorner [2], Syed Ghulam Rahman,[1] Magnus Helgesson [1], Torbjörn Ivert,[3] Ellenor Mittendorfer-Rutz[1]

For numbered affiliations see end of article.

**Correspondence to**
Dr Mo Wang; mo.wang@ki.se

## ABSTRACT

**Objectives** Research covering a wide range of risk factors related to the prognosis during the first year after an acute myocardial infarction (AMI) is insufficient. This study aimed to investigate whether sociodemographic, labour market marginalisation and medical characteristics before/at AMI were associated with subsequent reinfarction and all-cause mortality.

**Design** Population-based cohort study.

**Participants** The cohort included 15 069 individuals aged 25–64 years who had a first AMI during 2008–2010.

**Primary and secondary outcome measures** The outcome measures consisted of reinfarction and all-cause mortality within 1 year following an AMI, which were estimated by univariate and multivariable HRs and 95% CIs by Cox regression.

**Results** Sociodemographic characteristics such as lower education showed a 1.1-fold and 1.3-fold higher risk for reinfarction and mortality, respectively. Older age was associated with a higher risk of mortality while being born in non-European countries showed a lower risk of mortality. Labour market marginalisation such as previous long-term work disability was associated with a twofold higher risk of mortality. Regarding medical characteristics, ST-elevation myocardial infarction was predictive for reinfarction (HR: 1.14, 95% CI: 1.07 to 1.21) and all-cause mortality (HR: 3.80, 95% CI: 3.08 to 4.68). Moreover, diabetes mellitus, renal insufficiency, stroke, cancer and mental disorders were associated with a higher risk of mortality (range of HRs: 1.24–2.59).

**Conclusions** Sociodemographic and medical risk factors were identified as risk factors for mortality and reinfarction after AMI, including older age, immigration status, somatic and mental comorbidities. Previous long-term work disability and infarction type provide useful information for predicting adverse outcomes after AMI during the first year, particularly for mortality.

### Strengths and limitations of this study

► This is a population-based cohort study on all patients with acute myocardial infarction from inpatient care.
► The Swedish national-wide register data have high quality, which reduces the risk of recall bias regarding exposure and outcome.
► Despite a wide range of risk factors that have been examined, some potential for residual confounding by unmeasured factors remains.
► There is no available information on sick-leave spells that are shorter than 14 days among employed individuals.

## INTRODUCTION

Acute myocardial infarction (AMI) is the leading cause of mortality worldwide and reinfarction is common, ranging from 8% to 20% in the first year.[1] Over the past decade, percutaneous coronary intervention (PCI) and medication have reduced mortality in patients with AMI.[2 3] Despite this progress, AMI remains a major cause of mortality and disability. For patients who survive a first AMI, postdischarge optimal medical management and healthy lifestyle are essential. Particularly, reinfarction and heart failure can occur after an AMI, influencing quality of life and increasing healthcare costs.[1 4 5] Knowledge of risk factors for reinfarction and mortality in the first year after an AMI could improve the ability of healthcare providers to reduce progression of disease as well as improve survival after AMI.

Previous studies have reported risk factors for reinfarction and mortality in patients with AMI, mainly focusing on events within the first month after discharge.[6] Sociodemographic characteristics such as older age, lower socioeconomic status, living alone and (co)morbidity (eg, diabetes mellitus, renal diseases, hypertension, unstable angina, stroke or transient ischaemic attack, cancer and depression) have been found to be associated with a higher risk of reinfarction and mortality after discharge.[6–10] None of these studies have taken into consideration risk factors for reinfarction or mortality in the mid-term that is, 1 year after hospital discharge. Moreover, currently there is little evidence related to crucial AMI-related characteristics such as type of coronary revascularisation and infarction. Here, studies are lacking which include a vast range of risk factors and are based on register data, which provide large study populations and guarantee practically no loss to follow-up.

Additionally, there is a lack of studies elucidating the associations between characteristics of labour market marginalisation and the risk of reinfarction and mortality among patients with AMI. In Sweden, more than 30 000 persons experience an AMI each year; of these, about 10 000 are below the age of 65.[11] This burden of disease may result in long-term work disability in the working age population.[12 13] To date, sickness absence (SA) is almost always prescribed as a rehabilitation strategy in healthcare services for patients with AMI.[14] Also, permanent work disability, disability pension (DP), is common in this patient group.[12] In a prior study, patterns of SA/DP before AMI provided crucial information for subsequent work disability.[15] To the best of our knowledge, this is the first study investigating labour market marginalisation measured in terms of trajectories of SA/DP and unemployment status as risk factors for reinfarction and mortality in patients with AMI.

### Aims
The study aimed to investigate to what extent sociodemographic, labour market marginalisation and medical (including AMI-related factors and comorbidities) characteristics before/at an AMI were associated with subsequent reinfarction and all-cause mortality during the first year.

## MATERIALS AND METHODS
### Study population
This is a nationwide register-based cohort study and the study population consisted of 16 983 individuals aged 25 to 64 years who had a first AMI during 2008–2010. A main diagnosis of AMI was ascertained from the inpatient care register and defined according to the International Classification of Diseases (ICD)-10 code of I21. This means that individuals with a previous main or side diagnosis of AMI in specialised healthcare from 1987 up to the hospital admission date for AMI during 2008–2010 were

excluded (n=1914). Altogether, there were 15 069 individuals included in the study.

### Registers
National register data were linked to the study population by using the unique personal identity number assigned to all Swedish inhabitants, including information for each individual up to 31 December 2013 from:
1. Statistics Sweden: sex, age, education, country of birth, type of living area, family situation, length of unemployment and year of emigration from the Longitudinal integration database for health insurance and labour market studies.
2. The Social Insurance Agency: SA/DP (date and grade) from Micro-data for analyses of social insurance.
3. The National Board of Health and Welfare: date and cause of diagnosis-specific inpatient and specialised outpatient care, and type of infarction and type of coronary revascularisation from the National Patient Register; date of death from the Cause of Death Register[16] and date, type and dose of prescription of dispensed psychiatric medication and antidiabetic medication from the National Prescribed Drug Register.

### Outcome measures
The outcome measures were reinfarction (ICD codes: I21) which was ascertained from the inpatient care, and all-cause mortality during the first year after AMI.

### Risk measures
Sociodemographic characteristics were recorded at the end of the year preceding AMI and comprised sex, age, education (low educational level (compulsory (≤9 years)), high school (10–12 years) and high educational level (university (>12 years))), country of birth, type of living area and family situation (table 1).

Labour market marginalisation characteristics included length of unemployment in the year preceding AMI and the trajectory groups of SA/DP during 3 years before and up to the AMI diagnosis (table 1). The trajectory groups of SA/DP were measured using the combined mean number of annual SA and DP net days before the AMI diagnosis. The total number of net days were then transformed to number of months with SA/DP.

Medical characteristics included AMI-related characteristics (type of infarction and type of coronary revascularisation) at inclusion and inpatient and specialised outpatient care due to any main or side diagnosis of somatic and mental comorbidities and medication which were measured from 3 years before until the AMI diagnosis. Type of infarction was classified as ST-elevation myocardial infarction (STEMI, ICD-codes: I21.0–I21.3), non-ST-elevation myocardial infarction (NSTEMI, ICD-codes: I21.4) or unspecified (ICD-codes: I21.9). Information on type of coronary revascularisation was categorised as: PCI (FNG00–FNG05), coronary artery bypass grafting

**Table 1** Descriptive statistics for all women (n=3673) and men (n=11 396) aged between 25 and 64 years with a diagnosis of AMI from inpatient care in 2008–2010 in Sweden (n=15 069)

| Characteristics of patients with AMI | All | | Women | | Men | | $\chi^2$ |
|---|---|---|---|---|---|---|---|
| | n 15 069 | % 100 | n 3673 | % 24.4 | n 11 396 | % 75.6 | (p value) |
| **Sociodemographic characteristics*** | | | | | | | |
| Age (years)† | | | | | | | |
| 25–45 | 1401 | 9.3 | 335 | 9.1 | 1066 | 9.4 | 15.1 (<0.001) |
| 46–55 | 4739 | 31.5 | 1065 | 29.0 | 3674 | 32.2 | |
| 56–64 | 8929 | 59.3 | 2273 | 61.9 | 6656 | 58.4 | |
| Education (years)†‡ | | | | | | | |
| Compulsory (≤9) | 4474 | 29.7 | 1040 | 28.3 | 3434 | 30.1 | 9.9 (<0.01) |
| High school (10–12) | 7435 | 49.3 | 1895 | 51.6 | 5540 | 48.6 | |
| University (>12) | 3160 | 21.0 | 738 | 20.1 | 2422 | 21.3 | |
| Country of birth†§ | | | | | | | |
| Sweden | 12 085 | 80.2 | 2991 | 81.4 | 9094 | 79.8 | 86.4 (<0.001) |
| Other Nordic countries | 860 | 5.7 | 282 | 7.7 | 578 | 5.1 | |
| Europe (except Nordic countries) | 440 | 2.9 | 116 | 3.2 | 324 | 2.8 | |
| Non-European countries | 1684 | 11.2 | 284 | 7.7 | 1400 | 12.3 | |
| Type of living area†¶ | | | | | | | |
| Big cities | 4566 | 30.3 | 1052 | 28.6 | 3514 | 30.8 | 6.5 (<0.05) |
| Medium-sized cities | 5347 | 35.5 | 1344 | 36.6 | 4003 | 35.1 | |
| Small towns/villages | 5156 | 34.2 | 1277 | 34.8 | 3879 | 34.0 | |
| Family situation†** | | | | | | | |
| Married†† living without children | 4880 | 32.4 | 1342 | 36.5 | 3538 | 31.1 | 235.8 (<0.001) |
| Married†† living with children | 4000 | 26.5 | 728 | 19.8 | 3272 | 28.7 | |
| Single‡‡ living without children | 5386 | 35.7 | 1271 | 34.6 | 4115 | 36.1 | |
| Single‡‡ living with children | 803 | 5.3 | 332 | 9.0 | 471 | 4.1 | |
| **Labour market marginalisation characteristics** | | | | | | | |
| Trajectory groups of SA/DP† from 3 years before up to inclusion | | | | | | | |
| Low increasing | 8048 | 53.4 | 1345 | 36.6 | 6703 | 58.8 | 705.4 (<0.001) |
| Low constant | 2714 | 18.0 | 709 | 19.3 | 2005 | 17.6 | |
| Middle increasing | 1420 | 9.4 | 455 | 12.4 | 965 | 8.5 | |
| High decreasing | 794 | 5.3 | 331 | 9.0 | 463 | 4.1 | |
| High constant | 2093 | 13.9 | 833 | 22.7 | 1260 | 11.1 | |
| Unemployment† in the year before inclusion | | | | | | | |
| No unemployment | 13 799 | 91.6 | 3420 | 93.1 | 10 379 | 91.1 | 15.0 (<0.001) |
| 1–180 days | 852 | 5.7 | 171 | 4.7 | 681 | 6.0 | |
| >180 days | 418 | 2.8 | 82 | 2.2 | 336 | 3.0 | |
| **AMI-related characteristics** | | | | | | | |
| Type of infarction†§§ at inclusion | | | | | | | |
| STEMI**** | 5260 | 34.9 | 1058 | 28.8 | 4202 | 36.9 | 84.7 (<0.001) |
| Non-STEMI†††† | 6704 | 44.5 | 1832 | 49.9 | 4872 | 42.8 | |
| Unspecified | 3105 | 20.6 | 783 | 21.3 | 2322 | 20.4 | |
| Coronary revascularisation characteristics†††† at inclusion | | | | | | | |
| Percutaneous coronary intervention | 10 364 | 68.8 | 2100 | 57.2 | 8264 | 72.5 | 353.9 (<0.001) |
| Coronary artery bypass grafting | 336 | 2.2 | 59 | 1.6 | 277 | 2.4 | |
| Others | 4369 | 29.0 | 1514 | 41.2 | 2855 | 25.1 | |

Continued

**Table 1** Continued

| Characteristics of patients with AMI | All | | Women | | Men | | χ² (p value) |
|---|---|---|---|---|---|---|---|
| | n 15 069 | % 100 | n 3673 | % 24.4 | n 11 396 | % 75.6 | |
| **Comorbidity characteristics§§** | | | | | | | |
| Somatic comorbidities‡‡‡ from 3 years before up to inclusion | | | | | | | |
| Musculoskeletal disorders† | 2299 | 15.3 | 741 | 20.2 | 1558 | 13.7 | 90.9 (<0.001) |
| Diabetes mellitus†§§§ | 2529 | 16.8 | 675 | 18.4 | 1854 | 16.3 | 8.8 (<0.01) |
| Renal insufficiency | 248 | 1.7 | 70 | 1.9 | 178 | 1.6 | 2.0 (0.15) |
| Hypertension† | 5110 | 33.9 | 1365 | 37.2 | 3745 | 32.9 | 22.9 (<0.001) |
| Stroke | 199 | 1.3 | 54 | 1.5 | 145 | 1.3 | 0.8 (0.36) |
| Cancer† | 933 | 6.2 | 303 | 8.3 | 630 | 5.5 | 35.4 (<0.001) |
| Other somatic disorders† | 10 107 | 67.1 | 2722 | 74.1 | 7385 | 64.8 | 108.9 (<0.001) |
| Mental comorbidities | | | | | | | |
| Common mental disorders†‡‡‡ from 3 years before up to inclusion | 791 | 5.3 | 287 | 7.8 | 504 | 4.4 | 64.2 (<0.001) |
| Other mental disorders‡‡‡ from 3 years before up to inclusion | 1331 | 8.8 | 328 | 8.9 | 1003 | 8.8 | 0.1 (0.81) |
| Psychiatric medication†¶¶¶ in the year before inclusion | 3231 | 21.4 | 1299 | 35.4 | 1932 | 16.7 | 559.1 (<0.001) |
| **Reinfarction and all-cause mortality during first year after AMI** | | | | | | | |
| Reinfarction | 5310 | 35.2 | 1276 | 34.7 | 4034 | 35.4 | 0.5 (0.47) |
| All-cause mortality† | 666 | 4.4 | 191 | 5.2 | 475 | 4.2 | 7.0 (<0.01) |

*Measured on 31 December of the year preceding AMI.
†Significant sex differences.
‡Missing data are considered compulsory education.
§Missing data are considered non-European countries.
¶Type of living area: big cities (Stockholm, Gothenburg and Malmö), medium-sized cities (cities with more than 90 000 inhabitants within 30 km distance from the centre of the city), small cities/villages/rural.
**Missing data are considered single living without children.
††Married includes all living with partner; cohabitant.
‡‡Single includes divorced, separated or widowed.
§§See Materials and methods section for the International Classification of Diseases, version 10 codes or the Anatomic Therapeutic Chemical classification system codes.
¶¶STEMI.
***N-STEMI.
†††See Materials and methods section for the Classification of Surgical Procedures.
‡‡‡Measured by main or side diagnosis in inpatient or specialised outpatient care.
§§§Additionally measured by prescribed antidiabetic medication.
¶¶¶Measured by antidepressants, anxiolytics and sedatives.
****STEMI: ST-elevation myocardial infarction.
††††Non-STEMI: Non-ST-elevation myocardial infarction.
AMI, acute myocardial infarction; DP, disability pension; N-STEMI, non-ST-elevation myocardial infarction; SA, sickness absence; STEMI, ST-elevation myocardial infarction.

(CABG) (FNA-FNF, FNG30, FNW96) and others (ie, other treatments/examinations or missing information).

Somatic comorbidities were categorised as musculoskeletal diagnoses (ICD codes: M00–99), renal insufficiency (ICD codes: N17–N19), stroke (ICD codes: I60, I61, I63, I64), hypertension (ICD codes: I10), cancer (ICD codes: C00–D48) and other somatic disorders (the other ICD codes except for mental diagnoses). The individuals with any specialised care due to diabetes mellitus or having any prescribed antidiabetic medication were coded according to ICD-codes: E10–E14 and the Anatomic Therapeutic Chemical classification system (ATC) code: A10. Mental comorbidities were grouped as CMDs (ie, depressive (ICD codes: F32–F33), anxiety (ICD codes: F40–F42) and stress-related disorders (ICD codes: F43)), and other mental disorders (ICD codes: F00–F31,

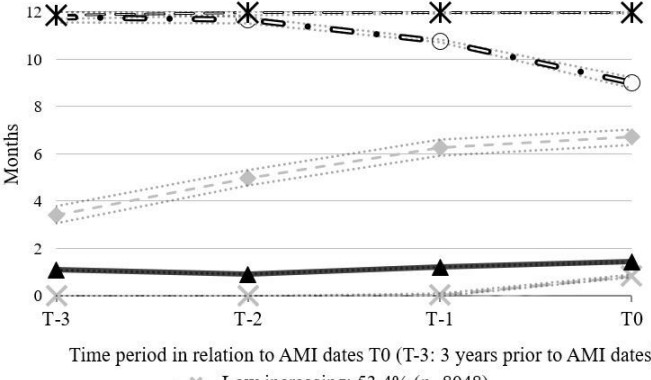

**Figure 1** Trajectory groups of sickness absence and disability pension (SA/DP) months before the hospital admission date for acute myocardial infarction (AMI) in 2008–2010 (T0) and percentages of individuals in each trajectory group (n=15 069). The dotted lines represent 95% CIs.

F34–F39 and F44–F99). Moreover, prescribed psychiatric medication during the year preceding the AMI diagnosis was included as mental comorbidities. Psychiatric medication was measured by any antidepressants, anxiolytics and sedatives following the ATC codes N06A, N05B and N05C, respectively.

## Statistical analyses

We used group-based trajectory modelling to estimate groups of SA/DP trajectories during the 3-year period before AMI. This method has been described elsewhere.[15 17] Five groups were selected as the best fitting model for patients with AMI. An annual time scale was used in the study, where T0 represents the first hospital admission date due to AMI and T-3 represents the 3 years before the first AMI diagnosis (figure 1). The five trajectory groups were named according to the patterns of each group: 'Low increasing', 'Low constant', 'Middle increasing', 'High decreasing' and 'High constant'.

$\chi^2$ tests were used to estimate potential sex differences regarding all the examined characteristics among patients with AMI. HRs and 95% CIs for reinfarction and all-cause mortality were calculated using Cox regression. The proportional hazards assumption was tested and met. Follow-up time started from the first hospital admission date due to AMI diagnosis until the events (reinfarction or all-cause mortality), emigration to a foreign country or the end of the first year after AMI, whichever came first. Mean follow-up time for reinfarction and all-cause mortality was 117 days (SD 120) and 177 days (SD 109), respectively. Interaction analyses were performed for sex and age; however, no interaction effects were found. We also carried out a sensitivity analysis with mortality due to cardiovascular diseases as outcome measure (online supplementary table). Analyses were adjusted for all risk measures in the multivariate model

(mental comorbidities were not mutually adjusted). Data processing was performed using SAS V.9.4.

## Patient and public involvement

There was no patient involvement in this study.

## RESULTS

Table 1 shows descriptive analysis for patients with a first AMI during 2008–2010. Of all, there were 3673 women (24.4%). The majority of the study population was older (56–64 years, 59.3%), born in Sweden (80.2%), belonged to the low increasing SA/DP trajectory group (53.4%) (figure 1), were not unemployed before inclusion (91.6%), received PCI at inclusion (68.8%), had other somatic disorders (67.1%) and did not have mental comorbidities. Reinfarction and all-cause mortality during the first year represented 35.2% and 4.4% of the study population, respectively. Furthermore, sex differences were significant for various factors. For example, with respect to labour market marginalisation characteristics, the 'Low increasing' SA/DP group comprised more men (58.8% vs 36.6%), while the 'High constant' SA/DP group was more common for women (22.7% vs 11.1%). Moreover, more men had a STEMI (36.9% vs 28.8%) and received a PCI (72.5% vs 57.2%) compared with women while more women had comorbidities compared with men.

### Reinfarction

In the univariate analyses, higher risks of reinfarction were found in those with lower education and living in small towns/villages (data not shown). In contrast, those born in non-Nordic European countries and those living in medium-sized cities had lower risks of subsequent reinfarction during the first year. Moreover, a higher risk of reinfarction was observed among those with STEMI compared with non-STEMI as well as among those treated with CABG compared with PCI (HR: 2.43; 95% CI: 2.14 to 2.75) (table 2).

In the final model, lower educational level and living in small towns/villages were associated with a higher risk of reinfarction while living in medium-sized cities, and being single living without children at home showed lower risk of reinfarction. With regard to AMI-related characteristics, patients with STEMI and CABG had a higher risk of reinfarction (table 2).

### All-cause mortality

In the multivariable model, we found that older age, lower level of education, being married/single living without children at home and belonging to the 'High constant' SA/DP trajectory group were risk factors for all-cause mortality during the first year after AMI. Those born in non-European countries and those belonging to the 'Low constant 'and 'High decreasing' SA/DP trajectory groups were associated with a lower risk of all-cause mortality. STEMI compared with non-STEMI was associated with a fourfold higher risk of all-cause mortality following AMI. Moreover, a higher risk of all-cause mortality was found in patients with diabetes mellitus, renal insufficiency, stroke, cancer and other somatic disorders compared with those

**Table 2** Adjusted HRs and 95% CIs for reinfarction in individuals with a diagnosis of AMI from inpatient care in 2008–2010 in Sweden (n=15 069) during the first year after AMI

| Characteristics of patients with AMI | Reinfarction n (%) | Model 1* HR (95% CI) | Model 2† | Model 3‡ |
|---|---|---|---|---|
| **Sociodemographic characteristics§** | | | | |
| Sex | | | | |
| Men | 4034 (35.4) | 1 | 1 | 1 |
| Women | 1276 (34.7) | 0.97 (0.91 to 1.04) | 0.98 (0.91 to 1.04) | 1.03 (0.97 to 1.11) |
| Age (years) | | | | |
| 25–45 | 482 (34.4) | 1 | 1 | 1 |
| 46–55 | 1679 (35.4) | 1.02 (0.92 to 1.13) | 1.02 (0.92 to 1.13) | 1.01 (0.91 to 1.12) |
| 56–64 | 3149 (35.3) | 0.98 (0.89 to 1.09) | 0.99 (0.90 to 1.10) | 0.99 (0.90 to 1.10) |
| Education (years)¶ | | | | |
| Compulsory (≤9) | 1630 (36.4) | 1.13 (1.05 to 1.23) | 1.13 (1.04 to 1.23) | 1.12 (1.04 to 1.22) |
| High school (10–12) | 2672 (35.9) | 1.11 (1.03 to 1.19) | 1.11 (1.03 to 1.19) | 1.10 (1.02 to 1.18) |
| University (>12) | 1008 (31.9) | 1 | 1 | 1 |
| Country of birth** | | | | |
| Sweden | 4367 (36.1) | 1 | 1 | 1 |
| Other Nordic countries | 309 (35.9) | 1.01 (0.90 to 1.13) | 1.01 (0.90 to 1.13) | 1.02 (0.91 to 1.14) |
| Europe (except Nordic countries) | 124 (28.2) | 0.83 (0.69 to 0.99) | 0.83 (0.69 to 0.99) | 0.83 (0.70 to 1.00) |
| Non-European countries | 510 (30.3) | 0.91 (0.82 to 1.00) | 0.91 (0.82 to 1.00) | 0.91 (0.82 to 1.00) |
| Type of living area†† | | | | |
| Big cities | 1368 (30.0) | 1 | 1 | 1 |
| Medium-sized cities | 1428 (26.7) | 0.84 (0.78 to 0.91) | 0.84 (0.78 to 0.91) | 0.84 (0.78 to 0.91) |
| Small towns/villages | 2514 (48.8) | 1.81 (1.69 to 1.94) | 1.81 (1.69 to 1.94) | 1.83 (1.71 to 1.96) |
| Family situation‡‡ | | | | |
| Married§§ living without children | 1782 (36.5) | 0.97 (0.90 to 1.05) | 0.97 (0.90 to 1.05) | 0.97 (0.90 to 1.05) |
| Married§§ living with children | 1439 (36.0) | 1 | 1 | 1 |
| Single¶¶ living without children | 1812 (33.6) | 0.89 (0.83 to 0.96) | 0.89 (0.83 to 0.96) | 0.89 (0.83 to 0.96) |
| Single¶¶ living with children | 277 (34.5) | 0.96 (0.84 to 1.09) | 0.96 (0.84 to 1.09) | 0.96 (0.84 to 1.10) |
| **Labour market marginalisation characteristics** | | | | |
| Trajectory groups of SA/DP from 3 years before up to inclusion | | | | |
| Low increasing | 2827 (35.1) | 1 | 1 | 1 |
| Low constant | 951 (35.0) | 0.96 (0.89 to 1.04) | 0.96 (0.89 to 1.04) | 0.97 (0.90 to 1.04) |
| Middle increasing | 501 (35.3) | 0.95 (0.87 to 1.06) | 0.96 (0.87 to 1.05) | 0.99 (0.89 to 1.09) |
| High decreasing | 295 (37.2) | 1.02 (0.90 to 1.16) | 1.02 (0.90 to 1.16) | 1.08 (0.95 to 1.23) |
| High constant | 736 (35.2) | 1.00 (0.92 to 1.09) | 1.00 (0.91 to 1.09) | 1.06 (0.97 to 1.17) |
| Unemployment in the year before inclusion | | | | |
| No unemployment | 4850 (35.2) | 1 | 1 | 1 |
| 1–180 days | 310 (36.4) | 1.07 (0.95 to 1.20) | 1.07 (0.95 to 1.20) | 1.06 (0.94 to 1.19) |
| >180 days | 150 (35.9) | 1.08 (0.92 to 1.27) | 1.08 (0.92 to 1.27) | 1.07 (0.91 to 1.26) |
| **AMI-related characteristics** | | | | |
| Type of infarction*** at inclusion | | | | |
| STEMI‡‡‡‡ | 1945 (37.0) | 1.19 (1.12 to 1.27) | 1.19 (1.12 to 1.27) | 1.14 (1.07 to 1.21) |
| Non-STEMI§§§§ | 2311 (34.5) | 1 | 1 | 1 |
| Unspecified | 1054 (34.0) | 1.02 (0.95 to 1.10) | 1.02 (0.95 to 1.10) | 1.05 (0.98 to 1.13) |

Continued

**Table 2**  Continued

| Characteristics of patients with AMI | Reinfarction<br>n (%) | Model 1*<br>HR (95% CI) | Model 2† | Model 3‡ |
|---|---|---|---|---|
| Coronary revascularisation characteristics§§§ at inclusion | | | | |
| Percutaneous coronary intervention | 3772 (36.4) | 1 | 1 | 1 |
| Coronary artery bypass grafting | 267 (79.5) | 2.31 (2.04 to 2.62) | 2.32 (2.05 to 2.63) | 2.41 (2.13 to 2.74) |
| Others | 1271 (29.1) | 0.71 (0.67 to 0.76) | 0.71 (0.67 to 0.76) | 0.74 (0.69 to 0.79) |
| **Comorbidity characteristics*** | | | | |
| Somatic comorbidities¶¶¶ from 3 years before up to inclusion | | | | |
| Musculoskeletal disorders | 835 (36.3) | 1.04 (0.96 to 1.12) | 1.04 (0.96 to 1.12) | 1.05 (0.98 to 1.14) |
| Diabetes mellitus**** | 827 (32.7) | 0.87 (0.81 to 0.94) | 0.87 (0.81 to 0.94) | 0.91 (0.84 to 0.98) |
| Renal insufficiency | 73 (29.4) | 0.74 (0.58 to 0.93) | 0.74 (0.59 to 0.94) | 0.84 (0.66 to 1.06) |
| Hypertension | 1741 (34.1) | 0.92 (0.87 to 0.98) | 0.93 (0.87 to 0.98) | 0.95 (0.89 to 1.00) |
| Stroke | 59 (29.7) | 0.76 (0.59 to 0.99) | 0.77 (0.59 to 0.99) | 0.81 (0.63 to 1.05) |
| Cancer | 300 (32.2) | 0.89 (0.79 to 1.00) | 0.90 (0.80 to 1.01) | 0.94 (0.83 to 1.06) |
| Other somatic disorders | 3507 (34.7) | 0.92 (0.87 to 0.97) | 0.92 (0.87 to 0.97) | 0.95 (0.90 to 1.01) |
| Mental comorbidities | | | | |
| Common mental disorders¶¶¶ from 3 years before up to inclusion | 300 (37.9) | 1.12 (1.00 to 1.27) | – | 1.13 (1.00 to 1.28) |
| Other mental disorders¶¶¶ from 3 years before up to inclusion | 474 (35.6) | 1.06 (0.96 to 1.17) | – | 1.05 (0.96 to 1.16) |
| Psychiatric medication†††† in the year before inclusion | 1101 (34.1) | 0.96 (0.90 to 1.04) | – | 1.00 (0.93 to 1.07) |

*Adjusted for sex, age, educational level, country of birth, type of living area, family situation, trajectory groups of SA/DP and previous unemployment.

†Adjusted for sex, age, educational level, country of birth, type of living area, family situation, trajectory groups of SA/DP and previous unemployment, inpatient and specialised outpatient care due to common mental disorders and other mental disorders, and psychiatric medications; mental comorbidities were not mutually controlled.

‡Adjusted for sex, age, educational level, country of birth, type of living area, family situation, trajectory groups of SA/DP and previous unemployment, inpatient and specialised outpatient care due to common mental disorders and other mental disorders, and psychiatric medications, type of infarction, type of coronary revascularisation, musculoskeletal disorders, diabetes mellitus, renal insufficiency, hypertension, stroke, cancer and other somatic disorders; mental comorbidities were not mutually controlled.

§Measured on 31 December of the year preceding AMI.

¶Missing data are considered compulsory education.

**Missing data are considered non-European countries.

††Type of living area: big cities (Stockholm, Gothenburg and Malmö), medium-sized cities (cities with more than 90 000 inhabitants within 30 km distance from the centre of the city), small cities/villages/rural.

‡‡Missing data are considered single living without children.

§§Married includes all living with partner; cohabitant.

¶¶Single includes divorced, separated or widowed.

***See Materials and methods section for the International Classification of Diseases, version 10 codes or the Anatomic Therapeutic Chemical classification system codes.

†††STEMI.

‡‡‡Non-STEMI.

§§§See Materials and methods section for the Classification of Surgical Procedures.

¶¶¶Measured by main or side diagnosis in inpatient or specialised outpatient care.

****Additionally measured by prescribed antidiabetic medication.

††††Measured by antidepressants, anxiolytics and sedatives.

‡‡‡‡STEMI: ST-elevation myocardial infarction.

§§§§Non-STEMI: Non-ST-elevation myocardial infarction.

AMI, acute myocardial infarction; DP, disability pension; N-STEMI, non-ST-elevation myocardial infarction; SA, sickness absence; STEMI, ST-elevation myocardial infarction.

without such comorbidities. Other mental disorders besides CMDs and psychiatric medication were significantly associated with subsequent all-cause mortality (table 3). The sensitivity analysis with mortality due to cardiovascular diseases as the outcome showed similar

**Table 3** Adjusted HRs and 95% CIs for all-cause mortality in individuals with a diagnosis of AMI from inpatient care in 2008–2010 in Sweden (n=15 069) during the first year after AMI

| Characteristics of patients with AMI | Mortality n (%) | Model 1* HR (95% CI) | Model 2† | Model 3‡ |
|---|---|---|---|---|
| **Sociodemographic characteristics§** | | | | |
| Sex | | | | |
| Men | 475 (4.2) | 1 | 1 | 1 |
| Women | 191 (5.2) | 0.95 (0.80 to 1.14) | 0.92 (0.78 to 1.10) | 0.94 (0.78 to 1.12) |
| Age (years) | | | | |
| 25–45 | 35 (2.5) | 1 | 1 | 1 |
| 46–55 | 137 (2.9) | 0.98 (0.67 to 1.42) | 0.97 (0.66 to 1.40) | 1.04 (0.71 to 1.51) |
| 56–64 | 494 (5.5) | 1.76 (1.23 to 2.50) | 1.74 (1.22 to 2.48) | 1.82 (1.27 to 2.60) |
| Education (years)¶ | | | | |
| Compulsory (≤9) | 255 (5.7) | 1.28 (1.02 to 1.62) | 1.29 (1.02 to 1.63) | 1.29 (1.02 to 1.62) |
| High school (10–12) | 305 (4.1) | 1.06 (0.85 to 1.33) | 1.06 (0.84 to 1.32) | 1.05 (0.84 to 1.31) |
| University (>12) | 106 (3.4) | 1 | 1 | 1 |
| Country of birth** | | | | |
| Sweden | 550 (4.6) | 1 | 1 | 1 |
| Other Nordic countries | 52 (6.1) | 1.02 (0.77 to 1.36) | 1.03 (0.78 to 1.38) | 1.22 (0.91 to 1.62) |
| Europe (except Nordic countries) | 16 (3.6) | 0.70 (0.42 to 1.15) | 0.73 (0.44 to 1.20) | 0.80 (0.48 to 1.32) |
| Non-European countries | 48 (2.9) | 0.58 (0.43 to 0.80) | 0.59 (0.44 to 0.81) | 0.62 (0.45 to 0.84) |
| Type of living area†† | | | | |
| Big cities | 214 (4.7) | 1 | 1 | 1 |
| Medium-sized cities | 224 (4.2) | 0.84 (0.69 to 1.02) | 0.84 (0.69 to 1.01) | 0.88 (0.72 to 1.06) |
| Small towns/villages | 228 (4.4) | 0.84 (0.69 to 1.01) | 0.85 (0.70 to 1.03) | 0.86 (0.71 to 1.05) |
| Family situation‡‡ | | | | |
| Married§§ living without children | 208 (4.3) | 1.31 (1.02 to 1.70) | 1.30 (1.01 to 1.68) | 1.36 (1.05 to 1.76) |
| Married§§ living with children | 94 (2.4) | 1 | 1 | 1 |
| Single¶¶ living without children | 334 (6.2) | 1.75 (1.38 to 2.22) | 1.69 (1.33 to 2.14) | 1.73 (1.36 to 2.20) |
| Single¶¶ living with children | 30 (3.7) | 1.28 (0.84 to 1.94) | 1.25 (0.83 to 1.90) | 1.28 (0.84 to 1.94) |
| **Labour market marginalisation characteristics** | | | | |
| Trajectory groups of SA/DP from 3 years before up to inclusion | | | | |
| Low increasing | 243 (3.0) | 1 | 1 | 1 |
| Low constant | 69 (2.5) | 0.82 (0.63 to 1.07) | 0.79 (0.60 to 1.03) | 0.68 (0.52 to 0.89) |
| Middle increasing | 71 (5.0) | 1.60 (1.22 to 2.09) | 1.47 (1.12 to 1.93) | 1.02 (0.78 to 1.35) |
| High decreasing | 18 (2.3) | 0.59 (0.36 to 0.95) | 0.52 (0.32 to 0.85) | 0.33 (0.20 to 0.54) |
| High constant | 265 (12.7) | 3.94 (3.26 to 4.76) | 3.45 (2.81 to 4.23) | 2.16 (1.75 to 2.70) |
| Unemployment in the year before inclusion | | | | |
| No unemployment | 622 (4.5) | 1 | 1 | 1 |
| 1–180 days | 27 (3.2) | 1.07 (0.72 to 1.59) | 1.05 (0.71 to 1.56) | 1.04 (0.70 to 1.55) |
| >180 days | 17 (4.1) | 1.31 (0.81 to 2.14) | 1.31 (0.81 to 2.14) | 1.28 (0.79 to 2.09) |
| **AMI-related characteristics** | | | | |
| Type of infarction*** at inclusion | | | | |
| STEMI‡‡‡‡ | 267 (5.1) | 2.45 (2.01 to 2.99) | 2.48 (2.03 to 3.03) | 3.80 (3.08 to 4.68) |
| Non-STEMI§§§§ | 153 (2.3) | 1 | 1 | 1 |
| Unspecified | 246 (7.9) | 3.45 (2.82 to 4.23) | 3.45 (2.82 to 4.22) | 2.97 (2.42 to 3.65) |
| **Coronary revascularisation characteristics§§§ at inclusion** | | | | |

Continued

**Table 3** Continued

| Characteristics of patients with AMI | Mortality<br>n (%) | Model 1*<br>HR (95% CI) | Model 2† | Model 3‡ |
|---|---|---|---|---|
| Percutaneous coronary intervention | 257 (2.5) | 1 | 1 | 1 |
| Coronary artery bypass grafting | 11 (3.3) | 1.26 (0.69 to 2.31) | 1.27 (0.70 to 2.33) | 1.65 (0.90 to 3.02) |
| Others | 398 (9.1) | 3.30 (2.81 to 3.87) | 3.25 (2.76 to 3.81) | 3.60 (3.03 to 4.28) |
| **Comorbidity characteristics*** | | | | |
| Somatic comorbidities¶¶¶ from 3 years before up to inclusion | | | | |
| Musculoskeletal disorders | 112 (4.9) | 0.90 (0.73 to 1.11) | 0.91 (0.74 to 1.12) | 1.00 (0.81 to 1.24) |
| Diabetes mellitus**** | 199 (7.9) | 1.73 (1.46 to 2.05) | 1.74 (1.47 to 2.06) | 1.70 (1.42 to 2.03) |
| Renal insufficiency | 60 (24.2) | 4.34 (3.31 to 5.71) | 4.29 (3.26 to 5.64) | 2.59 (1.95 to 3.45) |
| Hypertension | 198 (3.9) | 0.74 (0.62 to 0.87) | 0.74 (0.62 to 0.87) | 0.68 (0.57 to 0.81) |
| Stroke | 25 (12.6) | 1.96 (1.31 to 2.94) | 1.99 (1.33 to 2.98) | 1.63 (1.09 to 2.45) |
| Cancer | 102 (10.9) | 2.46 (1.99 to 3.05) | 2.45 (1.98 to 3.04) | 2.22 (1.78 to 2.75) |
| Other somatic disorders | 530 (5.2) | 1.62 (1.34 to 1.96) | 1.59 (1.31 to 1.92) | 1.46 (1.20 to 1.78) |
| Mental comorbidities | | | | |
| Common mental disorders¶¶¶ from 3 years before up to inclusion | 43 (5.4) | 0.80 (0.59 to 1.11) | – | 0.90 (0.66 to 1.24) |
| Other mental disorders¶¶¶ from 3 years before up to inclusion | 101 (7.6) | 1.40 (1.13 to 1.74) | – | 1.46 (1.17 to 1.82) |
| Psychiatric medication in the year before inclusion†††† | 241 (7.5) | 1.39 (1.16 to 1.66) | – | 1.24 (1.03 to 1.48) |

*Adjusted for sex, age, educational level, country of birth, type of living area, family situation, trajectory groups of SA/DP and previous unemployment.

†Adjusted for sex, age, educational level, country of birth, type of living area, family situation, trajectory groups of SA/DP and previous unemployment, inpatient and specialised outpatient care due to common mental disorders and other mental disorders, and psychiatric medications; mental comorbidities were not mutually controlled.

‡Adjusted for sex, age, educational level, country of birth, type of living area, family situation, trajectory groups of SA/DP and previous unemployment, inpatient and specialised outpatient care due to common mental disorders and other mental disorders, and psychiatric medications, type of infarction, type of coronary revascularisation, musculoskeletal disorders, diabetes mellitus, renal insufficiency, hypertension, stroke, cancer and other somatic disorders; mental comorbidities were not mutually controlled.

§Measured on 31 December of the year preceding AMI.

¶Missing data are considered compulsory education.

**Missing data are considered non-European countries.

††Type of living area: big cities (Stockholm, Gothenburg and Malmö), medium-sized cities (cities with more than 90 000 inhabitants within 30 km distance from the centre of the city), small cities/villages/rural.

‡‡Missing data are considered single living without children.

§§Married includes all living with partner; cohabitant.

¶¶Single includes divorced, separated or widowed.

***See Materials and methods section for the International Classification of Diseases, version 10 codes or the Anatomic Therapeutic Chemical classification system codes.

†††STEMI.

‡‡‡Non-STEMI.

§§§See Materials and methods section for the Classification of Surgical Procedures.

¶¶¶Measured by main or side diagnosis in inpatient or specialised outpatient care.

****Additionally measured by prescribed antidiabetic medication.

††††Measured by antidepressants, anxiolytics and sedatives.

‡‡‡‡STEMI: ST-elevation myocardial infarction.

§§§§Non-STEMI: Non-ST-elevation myocardial infarction.

AMI, acute myocardial infarction; DP, disability pension; N-STEMI, non-ST-elevation myocardial infarction; SA, sickness absence; STEMI, ST-elevation myocardial infarction.

results as for all-cause mortality (online supplementary table).

## DISCUSSION
### Sociodemographic and labour market marginalisation
Sociodemographic and labour market marginalisation were generally more associated with mortality than reinfarction in patients with AMI. For instance, results showed that a lower education level, which acts as a proxy of lower socioeconomic status, was associated with a less favourable prognosis regarding reinfarction (HR: 1.12) and all-cause mortality (HR: 1.29) during the first year after AMI. Previous studies have shown that patients with a lower educational level generally have a higher risk profile, primarily due to the presence of more risk factors such as smoking or the resistance of quitting smoking after AMI and comorbidities, leading to a worse health outcome.[18–20] After adjustment for comorbidities, we found that educational level remained an independent predictor of reinfarction and mortality, Still, one cannot rule out the possibility of unmeasured residual comorbidities that may be associated with reinfarction and all-cause mortality.

As expected, we observed that higher age was a strong predictor of all-cause mortality after AMI, which is in agreement with other studies.[21–23] Somewhat unexpectedly, older age was not associated with reinfarction during the first year. The different findings with respect to mortality and reinfarction may be driven by the comorbidities that were controlled for in the model, which are closely related with AMI and the association between age and all-cause mortality might be caused by other comorbidities. Interestingly, we found a higher risk of reinfarction for patients who were living in small towns/villages, while a lower risk of reinfarction was observed for those living in medium-sized cities compared with those living in big cities. This result might indicate diversities in healthcare in relation to different types of living area.[24]

Furthermore, patients with AMI born in non-European countries had a lower risk of all-cause mortality during the first year than patients born in Sweden. Recent research has shown a lower risk of mortality after AMI among South Asians compared with the host population.[25 26] Our finding may also reflect a 'healthy migrant effect', indicating a positive health selection of migrants who are able to overcome the obstacles of migration. Previous studies showed that migrants have revealed a lower risk of morbidity and mortality compared with natives.[27 28]

Compared with patients with AMI who were married and living with children at home, those who were married/single and living without children at home had a higher risk of all-cause mortality. Patients who live alone may have poor adherence to medication and follow-up recommendations, which might be associated with an unfavourable outcome. The few studies that have described the association between social support and prognosis in patients with coronary artery disease have had inconsistent definitions of measures of social support, leading to a wide variety of conclusions.[29] Therefore, the impact of family situation on reinfarction and all-cause mortality is open to speculation and warrants further investigation.

With regard to labour market marginalisation factors, the 'High constant' SA/DP trajectory group was associated with a 2.2-fold higher risk of all-cause mortality, even after controlling for confounders. As this group had around 12 months of SA/DP per annum before AMI, it is likely that this group consisted of a larger proportion of individuals with long-term SA or DP. This group may also have had a history of comorbidities before AMI, which in turn increases the risk of all-cause mortality. On the other hand, the 'Low constant' and 'High decreasing' SA/DP trajectory groups showed a lower risk of all-cause mortality after adjusting for comorbidities. Our study is the first to report that SA/DP trajectory groups can be used as risk factors for mortality in patients with AMI. Our findings also revealed that risk estimates of SA/DP trajectory groups were comparable to well-known risk factors such as diabetes mellitus and renal insufficiency. Therefore, more attention in clinical practice in relation to work disability factors in patients with AMI is necessary.

### Medical characteristics
Patients with comorbidities and STEMI had a higher risk of adverse outcomes, particularly for all-cause mortality, while those who underwent CABG had a higher risk of reinfarction. Indeed, STEMI is clinically associated with more serious medical conditions than non-STEMI.[30] With respect to coronary revascularisation, some studies have found that patients treated with PCI rather than CABG had fewer complications and a lower risk of mortality, particularly in the short term.[31 32]

In addition, a higher risk of all-cause mortality was observed among patients with mental comorbidities. Both biological and behavioural mechanisms have been suggested to explain the association between mental disorders and cardiovascular disease. Patients with mental disorders have been reported to have several cardiac symptoms.[33 34] Further, they tend to have poorer diets, reduced medication adherence and more stress.[35] Overall, mental disorders reduce the success of interventions targeting cardiovascular risk factor modification, leading to higher healthcare costs, poorer health outcomes and increased mortality rates.

### Strengths and limitations
The strengths of this study include the use of a population-based cohort design, which offers satisfactory statistical power for the analyses. The use of high-quality national register data also minimises the risk of recall bias regarding exposure and outcome.[36 37] There still might be misclassification and missing information in the register data. However, misclassification and missing information seems to be randomly distributed across the different exposure and outcome measures and this misclassification is assumed to be non-differential.

The high coverage of the register data also enabled us to identify all patients with AMI from inpatient care with subsequent reinfarction and mortality. We included only patients with AMI who were treated in inpatient care with more severe cardiac disease. This might explain the high incidence of reinfarction during the first year of the study. We also used an advanced method covering the inherent heterogeneity, group-based trajectory modelling, to investigate work disability patterns in the study. Moreover, we were able to examine a wide range of risk factors as well as adjust for relevant confounders. Still, there might be other factors than those studied here that are associated with reinfarction and mortality. Our registers did not include information of compliance to prescribed medication such as dual-antiplatelet therapy, smoking habits before and after AMI, rehabilitation measures and lifestyle changes.

Limitations of the study and considerations when interpreting our findings are acknowledged. In this study, we only included comorbidities recorded in inpatient and specialised outpatient care, but not those from primary care due to lack of data availability. While we adjusted for potential confounders that were particularly relevant for AMI, we acknowledge that there may be a wider range of comorbidities that we were unable to control for. Mental comorbidities were measured by including prescribed psychiatric medication data. For somatic comorbidities, we did not include an equivalent measure except for diabetes mellitus as there was no available information in the register data. With regard to SA, we did not have information on sick-leave spells that were less than 14 days among employed individuals. Thus, the number of SA days contributing to the combined number of SA/DP days might be underestimated.

## CONCLUSIONS

Several sociodemographic and comorbidity risk factors were generally associated more strongly with mortality than reinfarction in patients with AMI, including lower educational level, older age, immigration status, somatic and mental comorbidities. Previous long-term work disability and infarction type showed a higher risk for all-cause mortality after AMI during the first year.

**Author affiliations**
[1]Division of Insurance Medicine, Department of Clinical Neuroscience, Karolinska Institute, Stockholm, Sweden
[2]Department of Social and Preventive Medicine, Centre for Public Health, Medizinische Universitat Wien, Wien, Austria
[3]Department of Molecular Medicine and Surgery, Karolinska Institute, Stockholm, Sweden

**Contributors** EM-R, MV and MW conceived and designed the study. MW and EM-R were involved in the statistical analysis and drafted the manuscript. MW, MV, TED, SGR, MH, TI and EM-R contributed to the critical revision and approved the manuscript.

**Funding** This study was supported by the Swedish Research Council, grant no: 2015-02292.

**Competing interests** None declared.

**Patient consent for publication** Not required.

**Ethics approval** The study has been evaluated and approved by the Regional Ethical Review Board of Karolinska Institutet, Stockholm, Sweden (2007/762-31). The ethical review board approved the study and waived the requirement that informed consent of research subjects should be obtained.

**Provenance and peer review** Not commissioned; externally peer reviewed.

**Data availability statement** Data may be obtained from a third party and are not publicly available.

**ORCID iDs**
Mo Wang http://orcid.org/0000-0003-4036-3300
Thomas Ernst Dorner http://orcid.org/0000-0002-5218-1160
Magnus Helgesson http://orcid.org/0000-0002-7868-9712

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
