## [Reviewer comments · BMJ Open]

ARTICLE DETAILS

TITLE (PROVISIONAL)	Socio-demographic, labour market marginalisation, and medical characteristics as risk factors for re-infarction and mortality within one year after a first acute myocardial infarction-A register-based cohort study of a working age population in Sweden
AUTHORS	Wang, Mo; Vaez, Marjan; Dorner, Thomas; Rahman, Syed; Helgesson, Magnus; Ivert, Torbjörn; Mittendorfer-Rutz, Ellenor

VERSION 1 – REVIEW

REVIEWER	Christa Meisinger Helmholtz Zentrum München, German Research Center for Environmental Health, Germany
REVIEW RETURNED	02-Sep-2019

GENERAL COMMENTS	Based on 15069 patients aged 25 to 64 years who had a first acute myocardial infarction (AMI) during 2008-2010 in Sweden, it was investigated, whether socio-demographic, labour market marginalisation, and medical characteristics before/at AMI were associated with subsequent re-infarction and all-cause mortality within 1 year after AMI. For the analysis data from the Swedish national-wide inpatient-care register was used. It was found that socio-demographic (lower educational level, older age, immigration status) and medical risk factors (somatic and mental co-morbidities) were associated with mortality and re-infarction. I would make the following comments: Is there information on cause-specific mortality available? If yes, the authors should use cause-specific mortality (CVD-mortality, cancer-mortality, and all-cause mortality) as outcome and re-analyse the data. The authors state that the national-wide register data has high quality. Is it possible that encodings in the data are inaccurate or incorrect? How do such inaccuracies affect the results of the study? It is thinkable, that there are differences between men and women as well as between younger and older participants regarding the risk factors for re-infarction and mortality and thus, that the results differ due to sex and age. Did the authors conduct formal tests on interaction with sex and age? If yes, the results should be reported in the manuscript. If no, such tests should be carried out and if significant, stratified analyses should be conducted. The crude models do not necessarily need to be shown in the
---

	tables. The discussion section is rather long and should be shortened to about four pages.
REVIEWER	Željko Reiner University Hospital Center Zagreb, School of Medicine, University of Zagreb, Croatia
REVIEW RETURNED	05-Oct-2019
GENERAL COMMENTS	This is a very interesting manuscript. I have just some minor suggestions to the authors. In the Introduction, line 76 when mentioning the healthcare costs instead of ref. 1 I would suggest as more appropriate to cite 2 references: De Smedt D, et al. Cost-effectiveness of optimized adherence to prevention guidelines in European patients with coronary heart disease: Results from the EUROASPIRE IV survey. Int J Cardiol. 2018 Dec 1;272:20-25. AND De Smedt D, et al. Cost-effectiveness of optimizing prevention in patients with coronary heart disease: the EUROASPIRE III health economics project. Eur Heart J. 2012 Nov;33(22):2865-72. In Discussion on page 13, after the sentence starting in line 245 when discussing the influence of education I would suggest to cite the results of the big European survey: Bruthans J, et al EUROASPIRE IV investigators. Educational level and risk profile and risk control in patients with coronary heart disease. Eur J Prev Cardiol. 2016 May;23(8):881-90.

VERSION 1 – AUTHOR RESPONSE

Reviewer 1:

1. Is there information on cause-specific mortality available? If yes, the authors should use cause-specific mortality (CVD-mortality, cancer-mortality, and all-cause mortality) as outcome and re-analyse the data.

“Thank you for your comments. We have found that cancer mortality during the first year after the first acute myocardial infarction (AMI) is very rare and analyses regarding risk of cancer are underpowered. Same applies for other specific causes of mortality. Mortality due to cardiovascular disease (CVD) is an important outcome among patients with AMI and related analyses are sufficiently powered. We have therefore presented related results from CVD mortality in Supplementary Table 1. We found similar results as for all-cause mortality. These findings are indicated in the manuscript on page 10 and lines 193-194 and page 12 and lines 239-240.”

2. The authors state that the national-wide register data has high quality. Is it possible that encodings in the data are inaccurate or incorrect? How do such inaccuracies affect the results of the study?

“The quality of the Swedish registers is considered to be good. Previous validation studies showed high validity of Swedish register data on inpatient care diagnoses (Ludvigsson et al, 2011, Brooke et al, 2017). Moreover, the patient register has high sensitivity for most surgical procedures. To some extent there could of course be some coding errors in the data. However, misclassification and missing information are randomly distributed across the different outcome measures and this misclassification is assumed to be non-differential. We have added this discussion on page 16, lines 336-340.”

Ludvigsson JF, Andersson E, Ekbom A, Feychting M, Kim JL, Reuterwall C, et al. External review and validation of the Swedish national inpatient register. BMC public health. 2011;11:450.
 Brooke, H. L., Talbäck, M., Hörnblad, J., Johansson, L. A., Ludvigsson, J. F., Druid, H., Feychting, M., Ljung, R. (2017). The Swedish cause of death register. European Journal of Epidemiology, 32(9), 765-773. doi:10.1007/s10654-017-0316-1

3. It is thinkable, that there are differences between men and women as well as between younger and older participants regarding the risk factors for re-infarction and mortality and thus, that the results differ due to sex and age. Did the authors conduct formal tests on interaction with sex and age? If yes, the results should be reported in the manuscript. If no, such tests should be carried out and if significant, stratified analyses should be conducted.

“Thank you for pointing this out. Indeed, it is interesting to look at sex and age differences regarding the risk factors for re-infarction and mortality. Therefore, we have run interaction analyses for sex and age. These analyses revealed no interaction effects. Thus we do not present results from stratified analyses. We have added this information on page 10, lines 192-193.”

4. The crude models do not necessarily need to be shown in the tables.

“We have removed the crude models from the tables.”

5. The discussion section is rather long and should be shortened to about four pages.

“Thank you for this remark. We have shortened the discussion section to be shorter than four pages, please find the updated discussion on pages 13-18.”

Reviewer 2:

This is a very interesting manuscript. I have just some minor suggestions to the authors.
 In the Introduction, line 76 when mentioning the healthcare costs instead of ref. 1 I would suggest as more appropriate to cite 2 references: De Smedt D, et al. Cost-effectiveness of optimized adherence to prevention guidelines in European patients with coronary heart disease: Results from the EUROASPIRE IV survey. Int J Cardiol. 2018 Dec 1;272:20-25. AND
 De Smedt D, et al. Cost-effectiveness of optimizing prevention in patients with coronary heart disease: the EUROASPIRE III health economics project. Eur Heart J. 2012 Nov;33(22):2865-72.
 In Discussion on page 13, after the sentence starting in line 245 when discussing the influence of education I would suggest to cite the results of the big European survey: Bruthans J, et al EUROASPIRE IV investigators. Educational level and risk profile and risk control in patients with coronary heart disease. Eur J Prev Cardiol. 2016 May;23(8):881-90.

“Thank you for the suggested references. We have cited these references in the manuscript.”

VERSION 2 – REVIEW

REVIEWER	Christa Meisinger Ludwig-Maximilians-Universität München, Germany
REVIEW RETURNED	22-Nov-2019
GENERAL COMMENTS	The authors have satisfactorily answered to my comments and revised the manuscript accordingly. I have no further comments.